# Preventing Image Hallucination in Text-to-Image Generation through Factual Image Retrieval

**Youngsun Lim** , **Hyunjung Shim**

Kim Jaechul Graduate School of AI, KAIST

{youngsun_ai, kateshim}@kaist.ac.kr,

## Abstract

Text-to-image generation has shown remarkable progress due to the emergence of diffusion models. However, these models fail to reflect factual information and common sense inherent in the input text prompts, leading to the generation of factually inconsistent images. We define it as 'Image hallucination'. We categorize this problem into three types based on the study of hallucinations in language models and propose a methodology that uses factual images retrieved from external memory to generate realistic images. Depending on the target of the hallucination, we utilize either InstructPix2Pix or IP-Adapter, each method employing factual information from the retrieved factual images differently. This allows us to generate images that accurately reflect the facts and common sense contained in the input text prompts.

## 1 Introduction

Recently, text-to-image generation has made remarkable progress due to the emergence of diffusion models. However, many text-to-image diffusion models still do not properly understand the meaning and facts of input text prompts and generate images that are different from the real-world. For instance, the Statue of Liberty, completed in 1886, initially had a copper brown color because its surface was covered with copper. And over the decades, due to oxidation, the color gradually changed to its current blue-green color. However, when you enter the prompt 'The Statue of Liberty in 1890' into the recent text-to-image modelm such as Dall-E 3 [Betker *et al.*, 2023], only the turquoise Statue of Liberty is generated as shown in Figure 1.

The problem of generating inaccurate images spreads misinformation and misconceptions. This is a serious issue, especially when image generation models are used in fields where conveying facts is important, such as education or journalism. Furthermore, in the future, AI models trained on these incorrect images may develop serious biases. Most importantly, the reliability of AI models depends on their ability to provide accurate results; inaccurate generation reduces user trust in AI technology. Therefore, it is crucial that text-to-image generation depicts images based on facts.

Despite this importance, however, there is still little research solving these problems in text-to-image generation. Therefore, we define this problem as 'Image hallucination'. Image hallucination includes not only alignment problems between text prompts and generated images, but also the phenomenon of generating images that are different from reality. This is a higher concept than alignment because it requires understanding meaning and facts that are not included in the text prompt itself. In this paper, we focus on the problem of text-to-image generation failing to generate facts, excluding the alignment problem.

Because image hallucination is various, it is difficult to address all issues. Thus, we focus on the hallucinations that are judged to be representative based on [Huang *et al.*, 2023] by categorizing them into three types: Factual inconsistency caused by co-occurrence bias, factual inconsistency that cannot reflect time-shift information, and factual fabrication that produces counterfactual. We propose three types of prompts, incorrect generations, and modified images.

The problem can be solved by providing guidance to the image generation model using external memory of knowledge. Recently, retrieval-augmented language models have shown potential for advancement [Borgeaud *et al.*, 2022]. Given input text, such models retrieve relevant documents from external memory and generate fact-based answers. Recently, there have been studies that expanded retrieval and generation to both images and text and trained a multimodal generation model to use retrieval.

We develop this idea and propose a methodology that enhances an image generation model to generate fact-based images without training, by searching external factual images. Initially, an image is generated via an existing text-to-image model. Then, we search the text input prompt and retrieve N number of images in order of most relevance. Among them, the user selects the 'correct factual image' to be used as the guidance to eliminate image hallucination. Depending on the target where hallucination occurred, we propose two methods. (1) If hallucination occurs for the object or background, use InstructPix2Pix [Brooks *et al.*, 2023] to remove it. Referring to the way of generating instructions from InstructPix2Pix, the generated image and correct factual image are input into LLM (GPT-4) to generate instructions based on the difference between the two images. Then, input the instruction along with the initial generated image into the pre-trained

| DALL-E 3 | Retrieved image | Ours |
|:--:|:--:|:--:|

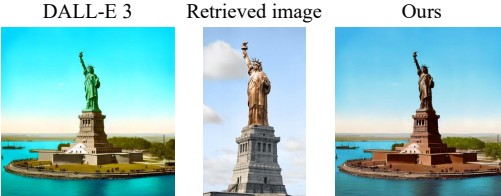 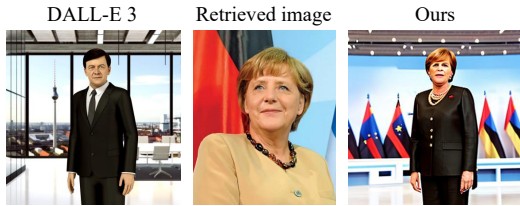

| DALL-E 3 | Retrieved image | Ours |
|:--:|:--:|:--:|

Input Prompt: The Statue of Liberty in 1890

Fact: **In 1890**, the Statue of Liberty had a **copper brown color**.

Input Prompt: The chancellor of Germany in 2015

Fact: The Chancellor of Germany in 2015 was **Angela Merkel, a 61-year-old woman at the time.**

Figure 1: Examples of image hallucination related with factual inconsistency.

InstructPix2Pix model to generate an image with hallucination removed. (2) If hallucination occurs in a person, use IP-Adapter [Ye *et al.*, 2023] to remove hallucination. Input text prompt and retrieved factual image are input into LLM (GPT-4) to generate a text prompt depicting the factual image. This generated prompt, correct factual image, and initial generated image are input into a pre-trained IP-Adapter model to remove hallucination present in the initial generated image using the image-to-image method.

Through this, we can eliminate hallucinations of existing text-to-image generation models without training costs. In addition, the user can interactively decide the retrieved image among the search results, thus reflecting the user's intention and making the edited image trustworthy.

## 2    Related Works

Text-to-image diffusion models have made significant progress but often struggle with complex prompts. Early methods used additional inputs like keypoints for better control [Yang *et al.*, 2023], while recent advancements leverage LLMs to manage layout directly [Wu *et al.*, 2023], improving prompt alignment. Diffusion models enable various image edits, from global styles to precise object manipulation [Hertz *et al.*, 2022], but often lack precision for detailed spatial adjustments. We address this by utilizing images from external sources to create fact-based images.

Some research focuses on generative models trained to retrieve in multimodal settings. Re-Imagen [Chen *et al.*, 2022b] generates images from retrieved images with text prompts, and MuRAG [Chen *et al.*, 2022a] generates language answers using retrieved images. Unlike these, our approach achieves similar effects without extensive training.

Various strategies exist to mitigate hallucination in LLMs. Data-related issues can be addressed by enhancing data quality [Lin *et al.*, 2021] and using better labeling techniques. For training-related hallucinations, improved model architectures and advanced regularization techniques [Liu *et al.*, 2024] are recommended. In this paper, we use high-quality, fact-based images to eliminate hallucinations.

## 3    Methods

### 3.1    Image Hallucination

We focus on the problem of the image not reflecting the common sense and facts behind the text, rather than the misalignment of the text and image. Accordingly, based on a paper analyzing hallucination in language model [Huang *et al.*, 2023], we address the representative image hallucination in which the generated image does not reflect the facts, and classify it into 3 categories. First, image hallucination can be classified into factual inconsistency and factual fabrication, and factual inconsistency is classified as being caused by co-occurrence bias and not reflecting time-shift information due to limited knowledge boundary.

Factual inconsistency refers to a situation where the output of a generation model contains facts based on real-world information, but is contradictory or inaccurate. Specifically, factual inconsistency caused by co-occurrence bias occurs because foundation models rely predominantly on co-occurrence patterns of certain data when the pre-training data of the model is rare. For instance, although the Statue of Liberty was originally copper brown, models generate it as its current bluish-green color due to the predominance of such data, ignoring its factual appearance.

Furthermore, there is a hallucination of factual inconsistency that occurs due to the inability to reflect time-shift information due to a knowledge boundary. Since the internal knowledge of the foundation model is not updated once trained, external information must be used to generate new updated knowledge over time. For example, current text-to-generation models cannot accurately generate images of presidents or chancellors from a specific time period.

Additionally, factual fabrication is a type of hallucination that generates cases that are unlikely or impossible when compared to real-world. For example, San Francisco rarely experiences snow in the winter, having only witnessed it three times since the 20th century. However, if you enter 'The Golden gate bridge in winter', which is a famous landmark in San Francisco, an image with a lot of snow is created.

### 3.2    Retrieval-augmented Factual Text-to-Image Generation

We propose two pipelines, as shown in Fig2, that utilizes retrieved factual images to apply real-world knowledge and common sense that the foundation model cannot reflect based on text prompts itself for image generation.

**Image Retrieval Interaction**

To obtain factual information about a given input text prompt, we use Google's Custom Search JSON API to retrieve im-

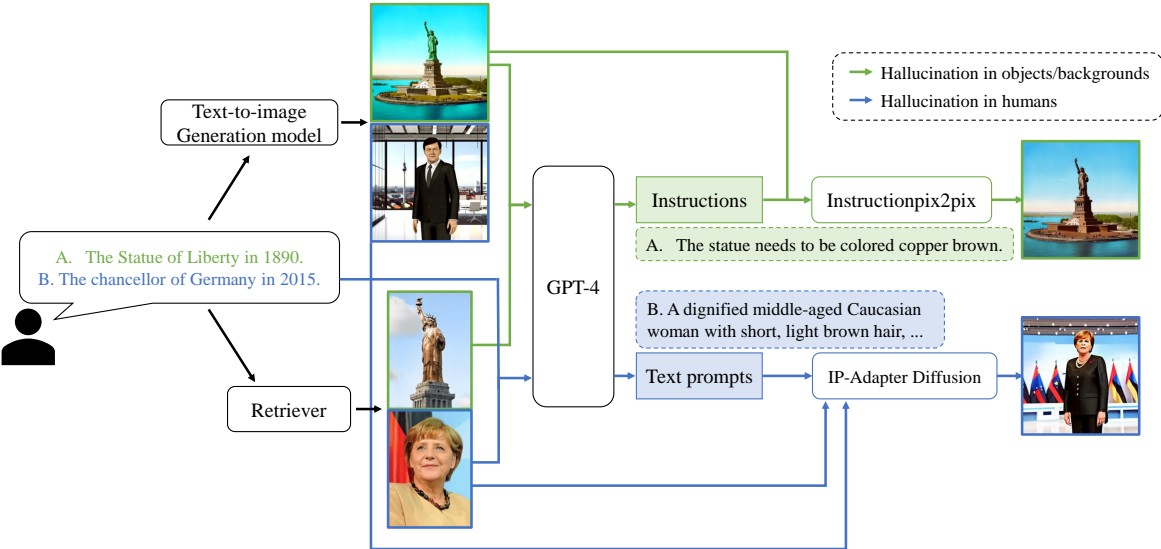

Figure 2: The overall pipeline indicating two different strategies for preventing image hallucination based on the subject of the hallucination.

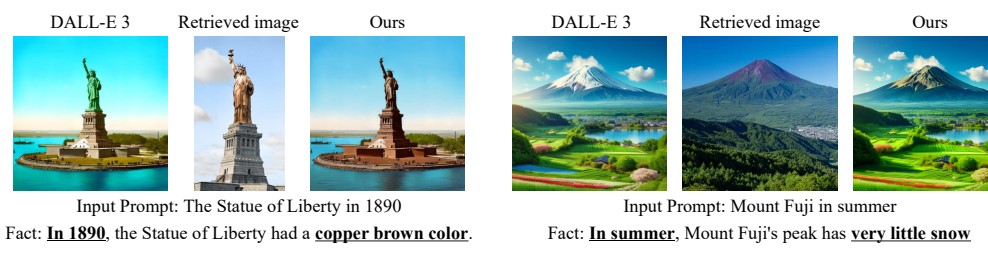

Input Prompt: The Statue of Liberty in 1890

Fact: **In 1890**, the Statue of Liberty had a **copper brown color**.

Instruction: "The statue needs to be colored copper brown."

Input Prompt: Mount Fuji in summer

Fact: **In summer**, Mount Fuji's peak has **very little snow**

Instruction: "Eliminate the snow on the mountain's summit."

Figure 3: Examples showing image hallucination due to factual inconsistency caused by co-occurrence bias, and images resolved by applying our methodology. Instructions are created and utilized using the input prompt and retrieved factual image.

ages for the prompt. Among the searched images, the user selects the image that best represents the factual information they wish to generate as the 'correct factual image.' The API can search up to 100 searches per day for free, after which additional fees apply. The number of images searched per prompt is a hyperparameter that varies depending on the individual's situation. If the desired image is not retrieved for a given prompt, increasing this number allows for a broader selection of candidate images.

**Overall pipeline**

We propose two methodologies that utilize the retrieved correct factual image, depending on the target of hallucination. First, if hallucination occurs on a specific object or background, we use the fact image to obtain instructions and use the InstructPix2Pix. InstructPix2Pix diffusion model, combining the knowledge of LLM (GPT-3) and text-to-image generation model (Stable Diffusion), edits images according to human instructions. The model's training dataset is constructed by entering the input caption into LLM (GPT-3) to obtain instructions and an edited caption, and entering the two captions into the diffusion model to obtain corresponding images. Referring to this, we input the initially generated image and the correct factual image into LLM (GPT-4) and

generate an instruction based on the difference between the two images. For example, because the biggest difference between the retrieved factual image for 'The Statue of Liberty in 1890' and the generated image is color, we input both images into GPT-4 to generate the instruction, 'The statue needs to be colored copper brown.' We input this instruction and the initially generated image together into the pre-trained InstructPix2Pix diffusion model to correct the hallucination in the initially generated image.

We propose two methods to utilize the retrieved factual image, depending on the hallucination target. First, for hallucinations on specific objects or backgrounds, we use the factual image to obtain instructions and employ InstructPix2Pix. InstructPix2Pix edits images based on human instructions. Its training data is created by generating instructions from captions using GPT-3 and obtaining images through the diffusion model. Based on it, we input the initial generated image and the factual image into GPT-4 to generate instructions based on their differences. For example, for 'The Statue of Liberty in 1890,' the instruction might be 'The statue needs to be colored copper brown'. The instruction, along with the initial image, is then input into the InstructPix2Pix to correct the hallucination.

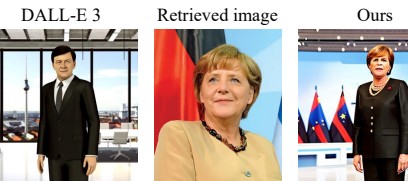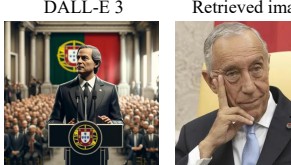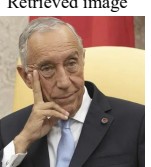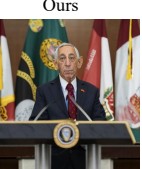

DALL-E 3      Retrieved image      Ours          DALL-E 3      Retrieved image      Ours

Input Prompt: The chancellor of Germany in 2015          Input Prompt: The president in Portugal on 30 May 2019

Fact: The Chancellor of Germany in 2015 was **Angela Merkel, a 61-year-old woman at the time.**      Fact: The President of Portugal on 30 May 2019 was **Marcelo Rebelo de Sousa, who was 70 years old at the time.**

Factual prompt: "A dignified middle-aged Caucasian woman with short, light brown hair, ..."      Factual prompt: "An elderly man with pronounced facial features and grey hair, ..."

Figure 4: Examples showing image hallucination due to factual inconsistency caused by failure to reflect time-shift information, and images resolved by applying our methodology. A factual prompt is generated and utilized using the input prompt and the retrieved image.

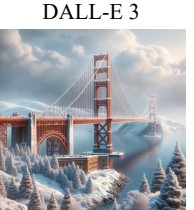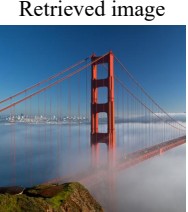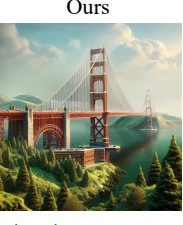

DALL-E 3      Retrieved image      Ours

Input Prompt: The Golden gate bridge in winter

Fact: **San Francisco rarely experiences** snow in winter

Instruction: "Remove the snow and replace the snowy landscape with a clear atmosphere."

Figure 5: Examples showing image hallucination due to factual fabrication, and images resolved by applying our methodology. Applying the same methodology as shown in Figure 3.

If hallucination occurs in complex subjects like a person (involving components such as face, hair, clothing, etc.), text prompts or text instructions alone may not be sufficient as generation conditions. Therefore, in order to input features that cannot be expressed in text, the correct factual image itself must be used as a prompt. For this purpose, we utilize an IP-Adapter that can input both image and text as prompts in combination with a pre-trained diffusion model. The IP-Adapter uses a decoupled cross-attention structure to process image and text features separately. Through this, you can create elaborate images using not only the text prompt but also the image prompt. To efficiently utilize these prompts, we input the input text prompt and the correct factual image into LLM (GPT-4) to create a prompt depicting the correct factual image. This newly generated text prompt and the correct factual image are then input into the IP-Adapter diffusion model along with the initially generated image to perform image-to-image editing, ultimately removing the hallucination from the initial generated image and editing it to reflect the features of the correct factual image.

## 4 Experiments and Results

We utilize DALL-E 3 as the model for initial image generation. We use GPT-4 as the LLM that generates instructions and prompts. The InstructPix2Pix and IP-Adapter models are pre-trained models based on Stable Diffusion v1.5.

Figure 3 illustrates hallucination due to factual inconsistency caused by co-occurrence bias and shows experimental results. The input prompt and the factual image (middle image in each example) are compared with DALL-E 3's initial output (left image in each example). The initial generation does not accurately reflect the facts. By inputting instructions derived from the differences between the initial and factual images into the InstructPix2Pix model, we obtain factually accurate images (right image in each example). The Statue of Liberty is correctly shown in its copper brown color as in 1890, and Mt. Fuji in the summer has a realistic appearance with almost all the snow on the top melting.

Figure 4 demonstrates the examples of hallucination related to factual inconsistencies that fail to account for time-shift information, along with experimental results. If the target where such hallucination occurred has complex and diverse factual information like a person, removing these inaccuracies through text is very challenging. Therefore, factual images from search results, like the ones in the middle of each example, are used as prompts. By entering the retrieved factual image and a prompt that well describes the image into the IP-Adapter model, images that reflect accurate factual information about individuals, as shown on the right of each example, can be obtained. Using our methodology, we can generate images that accurately depict Angela Merkel as the female Chancellor of Germany in 2015, and Marcelo Rebelo de Sousa as the President of Portugal in May 2019.

Figure 5 shows the hallucination and experimental results for factual fabrication. For generated images that do not properly reflect the fact that San Francisco rarely snows in winter, the same method used to address hallucinations caused by co-occurrence bias is applied. We obtain the instruction to remove all snow that causes hallucination in the initial generation from GPT-4. The instruction, along with the initially generated image, is input into the InstructPix2Pix model to generate an image containing the factual information of the retrieved factual image with all the snow removed.

## 5 Future works

We will further expand research to address more various hallucinations. We aim to resolve image hallucination more broadly and research metrics and benchmarks to measure it quantitatively and objectively.

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
