# OpenReview forum: "Preventing Image Hallucination in Text-to-Image Generation through Factual Image Retrieval"
_ijcai.org/IJCAI/2024/Workshop/TIDMwFM — IJCAI TIDMwFM 2024 Oral_

### Official Review · Reviewer_qqHx · 2024-06-19

**Rating:** 8
**Confidence:** 4

**Review:**

This paper presents a significant advancement in the domain of text-to-image generation. It tackles the pervasive issue of image hallucination, where generated images do not align with real-world facts, leading to misinformation. The authors categorize hallucinations into three types: co-occurrence bias, failure to reflect time-shift information, and factual fabrication, providing a structured understanding of the problem.

The proposed solution leverages external factual images retrieved via Google's Custom Search API, which are then used to guide the generation process. Two specific methodologies are detailed: InstructPix2Pix for correcting hallucinations in objects and backgrounds, and IP-Adapter for human subjects. This dual approach ensures that different types of hallucinations are addressed effectively.

One of the paper's strengths is its interactive component, allowing users to select the most accurate factual images from search results, thereby enhancing the trustworthiness and relevance of the generated images. The validation of the methodology through clear examples, such as the corrected coloration of the Statue of Liberty and the accurate depiction of historical figures, showcases its practical applicability and effectiveness.

Overall, the paper aligns well with the theme of "Trustworthy Interactive Decision-Making with Foundation Models Workshop" by focusing on enhancing the factual reliability of AI-generated content, a critical factor for building user trust in AI systems.

---

### Official Review · Reviewer_7CHF · 2024-06-21
**Review of the paper on preventing image hallucination**

**Rating:** 7
**Confidence:** 3

**Review:**

Summary: This paper addresses the problem of "image hallucination" in text-to-image generation models, where generated images fail to reflect factual information or common sense inherent in input prompts. The authors propose a methodology that uses factual images retrieved from external memory to guide the generation process, employing either InstructPix2Pix or IP-Adapter depending on the type of hallucination. The approach aims to generate images that accurately reflect facts and common sense without requiring model retraining.

Strengths and novelty: The paper introduces and categorizes the concept of "image hallucination," addressing an important issue in text-to-image generation. The approach allows for user input in selecting factual images, enhancing result reliability, matching with the theme of the workshop.

Weaknesses: The approach may introduce biases based on the retrieved images and LLM-generated instructions. Moreover, the multi-step process involving image retrieval, LLM prompt generation, and additional image generation may be computationally intensive.

Feedback to authors: To improve the paper further, the authors can consider the following: Discuss potential limitations and edge cases where the method might fail. Explore the scalability and computational requirements of the approach for large-scale applications.
Consider comparing the proposed method with other existing approaches for improving text-to-image generation accuracy.

---

### Official Review · Reviewer_D7Hk · 2024-06-21

**Rating:** 8
**Confidence:** 3

**Review:**

This study addresses the "image hallucination" issue in text-to-image generation models, where generated images fail to reflect factual information or common sense inherent in the input text prompts. A method is proposed to improve factual accuracy in generated images by using retrieved factual images as guidance.

Strengths:

1. The paper addresses an important and timely issue in text-to-image generation.

2. The proposed method is interesting in its use of retrieved factual images to guide image generation.

3. The method doesn't require retraining of large models, making it potentially more accessible and adaptable.

4. The examples provided clearly demonstrate the effectiveness of the approach.

Limitations and Potential Improvements:

1. The approach relies on the availability and accuracy of retrieved images, which may not always be guaranteed.

2. The method requires human intervention to select the correct factual image, which may limit scalability.

3. More discussion on the proposed method's computational costs and processing time would be beneficial.

---

### Decision · Program_Chairs · 2024-06-24

Accept (Oral)